# RIGGING THE LOTTERY: MAKING ALL TICKETS WINNERS

## ABSTRACT

Sparse neural networks have been shown to be more parameter and compute efficient compared to dense networks and in some cases they are even successfully used to decrease wall clock inference times. There is a large body of work on training dense networks to yield sparse networks for inference (Molchanov et al., 2017; Zhu & Gupta, 2018; Louizos et al., 2017; Li et al., 2016; Guo et al., 2016). This limits the size of the largest trainable sparse model to that of the largest trainable dense model. In this paper we introduce a method to train sparse neural networks with a fixed parameter count and a fixed computational cost throughout training, without sacrificing accuracy relative to existing dense-to-sparse training methods. Our method updates the topology of the network during training by using parameter magnitudes and infrequent gradient calculations. We show that this approach requires fewer floating-point operations (FLOPs) to achieve a given level of accuracy compared to prior techniques. We demonstrate state-of-the-art sparse training results with ResNet-50, MobileNet v1 and MobileNet v2 on the ImageNet-2012 dataset, WideResNets on the CIFAR-10 dataset and RNNs on the WikiText-103 dataset. Finally, we provide some insights into why allowing the topology to change during the optimization can overcome local minima encountered when the topology remains static.

## 1 INTRODUCTION

The parameter and floating point operation (FLOP) efficiency of sparse neural networks is now well demonstrated on a variety of problems (Han et al., 2015; Srinivas et al., 2017). Some work has even shown inference time speedups are possible on Recurrent Neural Networks (RNNs) (Kalchbrenner et al., 2018) and Convolutional Neural Networks (ConvNets) (Park et al., 2016). Currently, the most accurate sparse models are obtained with techniques that require, at a minimum, the cost of training a dense model in terms of memory and FLOPs (Zhu & Gupta, 2018; Guo et al., 2016), and sometimes significantly more (Molchanov et al., 2017). This paradigm has two main limitations:

1. The maximum size of sparse models is limited to the largest dense model that can be trained. Even if sparse models are more parameter efficient, we can't use pruning to train models that are larger and more accurate than the largest possible dense models.

2. It is inefficient. Large amounts of computation must be performed for parameters that are zero valued or that will be zero during inference.

Additionally, it remains unknown if the performance of the current best pruning algorithms are an upper bound on the quality of sparse models. Gale et al. (2019) found that three different dense-to-sparse training algorithms all achieve about the same sparsity / accuracy trade-off. However, this is far from conclusive proof that no better performance is possible. In this work we show the surprising result that dynamic sparse training, which includes the method we introduce below, *can find more accurate models* than the current best approaches to pruning initially dense networks. Importantly, our method does not change the FLOPs required to execute the model during training, allowing one to decide on a specific inference cost prior to training.

The Lottery Ticket Hypothesis (Frankle & Carbin, 2019) hypothesized that if we can find a sparse neural network with iterative pruning, then we can train that sparse network from scratch, to the

same level of accuracy, by *starting from the original initial conditions*. In this paper we introduce a new method for training sparse models without the need of a "lucky" initialization; for this reason, we call our method "The Rigged Lottery" or *RigL*[*]. We show that this method is:

- *Memory efficient*: It requires memory only proportional to the size of the sparse model. It never requires storing quantities that are the size of the dense model. This is in contrast to Dettmers & Zettlemoyer (2019) which requires storing the momentum for *all* parameters, even those that are zero valued.
- *Computationally efficient*: The amount of computation required to train the model is proportional to the number of nonzero parameters in the model.
- *Accurate*: The performance achieved by the method matches and sometimes *exceeds* the performance of pruning based approaches.

Our method works by infrequently using instantaneous gradient information to inform a re-wiring of the network. We show that this allows the optimization to escape local minima where it would otherwise become trapped if the sparsity pattern were to remain static. Crucially, as long as the full gradient information is needed less than every $\frac{1}{1-sparsity}$ iterations, then the overall work remains proportional to the model sparsity.

## 2 RELATED WORK

Research on finding sparse neural networks dates back decades; for example, at least to Thimm & Fiesler (1995) who concluded that pruning weights based on magnitude was a simple and powerful technique. Ström (1997) later introduced the idea of retraining the previously pruned network to increase accuracy. Han et al. (2016b) went further and introduced multiple rounds of magnitude pruning and retraining. This is, however, relatively inefficient, requiring ten rounds of retraining when removing $20\%$ of the connections to reach a final sparsity of $90\%$. To overcome this problem, Narang et al. (2017) introduced gradual pruning, where connections are slowly removed over the course of a single round of training. Zhu & Gupta (2018) refined the technique to minimize the amount of hyper-parameter selection required.

A diversity of approaches not based on magnitude based pruning have also been proposed. LeCun et al. (1990) and Hassibi & Stork (1993) are some early examples, but impractical for modern neural networks as they use information from the Hessian to prune a trained network. More recent work includes $L_0$ Regularization (Christos Louizos, 2018), Variational Dropout (Molchanov et al., 2017), Dynamic Network Surgery (Guo et al., 2016) and Sensitivity Driven Regularization (Tartaglione et al., 2018). Gale et al. (2019) examined magnitude pruning, $L_0$ Regularization and Variational Dropout and concluded that they all achieve about the same accuracy versus sparsity trade-off on ResNet-50 and Transformer architectures.

Training techniques that allow for sparsity throughout the entire training process were, to our knowledge, first introduced in Deep Rewiring (DeepR) (Bellec et al., 2017). In DeepR, the standard Stochastic Gradient Descent (SGD) optimizer is augmented with a random walk in parameter space. Additionally, connections have a pre-defined sign assigned at random; when the optimizer would normally flip the sign, the weight is set to 0 instead and new weights are activated at random.

Sparse Evolutionary Training (SET) (Mocanu et al., 2018) proposed a simpler scheme where weights are pruned according to the standard magnitude criterion used in pruning and are added back at random. The method is simple and achieves reasonable performance in practice. Dynamic Sparse Reparameterization (DSR) (Mostafa & Wang, 2019) introduced the idea of allowing the parameter budget to shift between different layers of the model, allowing for non-uniform sparsity. This allows the model to distribute parameters where they are most effective. Unfortunately, the models under consideration are mostly convolutional networks, so the result of this parameter reallocation (which is to decrease the sparsity of early layers and increase the sparsity of later layers) has the overall effect of increasing the FLOP count because the spatial size is largest at the beginning. Sparse Networks from Scratch (SNFS) (Dettmers & Zettlemoyer, 2019) introduces the idea of using the momentum of each parameter as the criterion to be used for growing weights and demonstrates it

---

[*]Pronounced "wriggle".

| Method | Drop | Grow | Selectable FLOPs | Space & FLOPs $\propto$ |
|---|---|---|---|---|
| SNIP | $min(\|\theta * \nabla_\theta L(\theta)\|)$ | none | yes | sparse |
| DeepR | stochastic | random | yes | sparse |
| SET | $min(\|\theta\|)$ | random | yes | sparse |
| DSR | $min(\|\theta\|)$ | random | no | sparse |
| SNFS | $min(\|\theta\|)$ | momentum | no | dense |
| RigL (ours) | $min(\|\theta\|)$ | gradient | yes | sparse |

Table 1: Comparison of different sparse training techniques. *Drop* and *Grow* columns correspond to the strategies used during the mask update. *Selectable FLOPs* is possible if the cost of training the model is fixed at the beginning of training.

leads to an improvement in test accuracy. Like DSR, they allow the sparsity of each layer to change and focus on a constant parameter, not FLOP, budget. Importantly, the method requires computing gradients and updating the momentum for *every* parameter in the model, even those that are zero, at *every* iteration. This can result in a significant amount of overall computation. Additionally, depending on the model and training setup, the required storage for the full momentum tensor could be prohibitive. Single-Shot Network Pruning (SNIP) (Lee et al., 2019) attempts to find an initial mask with one-shot pruning and uses the saliency score of parameters to decide which parameters to keep. After pruning training proceeds with this static sparse network. Properties of the different sparse training techniques are summarized in Table 1.

There has also been a line of work investigating the Lottery Ticket Hypothesis (Frankle & Carbin, 2019). Frankle et al. (2019) showed that the formulation must be weakened to apply to larger networks such as ResNet-50 (He et al., 2015). In large networks, instead of the original initialization, the values after thousands of optimization steps must be used for initialization. Zhou et al. (2019) showed that lottery tickets obtain non-random accuracies even before the training has started. Though the possibility of training sparse neural networks with a fixed sparsity mask using lottery tickets is intriguing, it remains unclear whether it is possible to generate such initializations – for both masks and parameters – *de novo*.

## 3 RIGGING THE LOTTERY

Our method, *RigL*, is illustrated in Figure 1. At regularly spaced intervals our method removes a fraction of connections based on weight magnitudes and activates new ones using instantaneous gradient information. After updating the connectivity, training continues with the updated network until the next update. The main parts of our algorithm, *Sparsity Distribution*, *Update Schedule*, *Drop Criterion*, *Grow Criterion*, and the various options we considered for each, are explained below. The improved performance of *RigL* is due to two reasons: the use of a new method for activating connections that is efficient and more effective than choosing at random, and the use of a natural extension to an existing method for distributing parameters statically among convolutional layers.

**(0) Notation.** Given a dataset $D$ with individual inputs $x_i$ and targets $y_i$, one can train a neural network to minimize the loss function $\sum_i L(f_\theta(x_i), y_i)$, where $f_\theta(x)$ is the neural network with

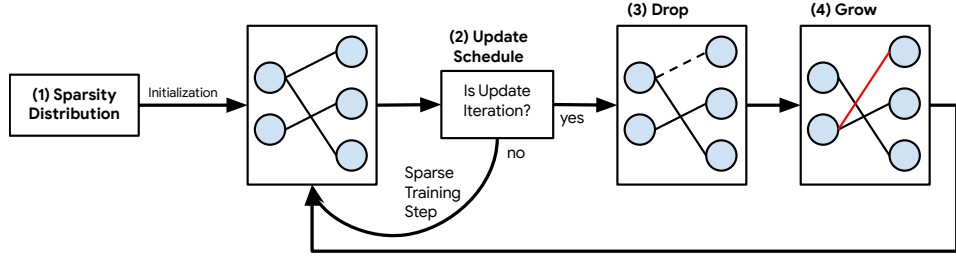

Figure 1: Dynamic sparse training aims to change connectivity during training to help out optimzation.

parameters $\theta$ of length $N$. The vector $\theta$ can be decomposed into parameters $\theta^l$, of length $N^l$, for each layer $l$. A sparse network keeps only a fraction $D \in (0, 1)$ of all connections, resulting in a **sparsity** of $S = 1 - D$. More precisely, denoting the sparsity of individual layers with $s^l$, the total parameter count of the sparse neural network satisfies $\sum_l (1 - s^l)N^l = (1 - S) * N$.

**(1) Sparsity Distribution.** There are many ways of distributing the non-zero weights across the layers while satisfying the equality above. We avoid re-allocating parameters between layers during the training process as it makes it difficult to target a specific final FLOP budget, which is important for many inference applications. We consider the following three strategies:

1. *Uniform:* The sparsity $s^l$ of each individual layer is the same as the total sparsity $S$. We keep the first layer dense ($s^0 = 0$), since it has negligible number of parameters.

2. *Erdős-Rényi:* As introduced in Mocanu et al. (2018), $s^l$ scales with $1 - \frac{n^{l-1}+n^l}{n^{l-1}*n^l}$, where $n^l$ denotes number of neurons at layer $l$. This enables the number of connections in a sparse layer to scale with the sum of the number of output and input channels.

3. *Erdős-Rényi-Kernel (ERK):* This method modifies the original Erdős-Rényi formulation by including the kernel dimensions in the scaling factors. In other words, the number of parameters of the sparse convolutional layers are scaled proportional to $1 - \frac{n^{l-1}+n^l+w^l+h^l}{n^{l-1}*n^l*w^l*h^l}$, where $w^l$ and $h^l$ are the width and the height of the $l$'th convolutional kernel. Sparsity of the fully connected layers scale as in the original Erdős-Rényi formulation. Similar to Erdős-Rényi, ERK allocates higher sparsities to the layers with more parameters while allocating lower sparsities to the smaller ones.

In all methods, the bias and batch-norm parameters are kept dense.

**(2) Update Schedule.** The update schedule is defined by the following parameters: (1) the number of iterations between sparse connectivity updates ($\Delta T$), (2) the iteration at which to stop updating the sparse connectivity ($T_{end}$), (3) the initial fraction of connections updated ($\alpha$) and (4) a function $f_{decay}$, invoked every $\Delta T$ iterations until $T_{end}$, possibly decaying the fraction of updated connections over time. For the latter we choose to use *cosine* annealing, as we find it slightly outperforms the other methods considered.

$$f_{decay}(t) = \frac{\alpha}{2}\left(1 + cos\left(\frac{t\pi}{T_{end}}\right)\right)$$

Alternatives to cosine annealing like a *constant* schedule and *inverse power* annealing are studied in the Appendix F.

**(3) Drop criterion.** Over the course of training, we drop the lowest magnitude weights according to the update schedule since they are expected to effect the training loss least. Specifically, we drop the connections given by
$TopK(-|\theta^l|, f_{decay}(t)(1 - s^l)N^l)^\ddagger$.

**(4) Grow criterion.** The novelty of our method lies in how we grow new connections. We grow the connections with highest magnitude gradients, $TopK_{w \notin \theta^l_{active}}(|grad(\theta^l)|, f_{decay}(t)(1 - s^l)N^l)$, where $\theta^l_{active}$ is the set of active connections after the drop step. Newly activated connections are initialized to zero and therefore don't effect the output of the network. However they are expected to receive gradients with high magnitudes in the next iteration and therefore reduce the loss fastest. This procedure can be applied to each layer in sequence and the dense gradients can be discarded immediately after selecting the top connections. If a layer is too large to materialize the full gradient with respect to the weights, then we can further reduce the memory requirements by performing an iterative calculation:

1. Initialize the set $TK = \{\}$.
2. Materialize a subset of size M of the full gradient, which we denote $G_{i:i+M}$.
3. Update $TK$ to contain the Top-K elements of $G_{i:i+M}$ concatenated with $TK$.
4. Repeat steps 1 through 3 until all of the gradients have been materialized. The final set $TK$ contains the connections we wish to grow.

---

$^\ddagger TopK(v, k)$ returns the indices and values of the top-$k$ elements of vector $v$.

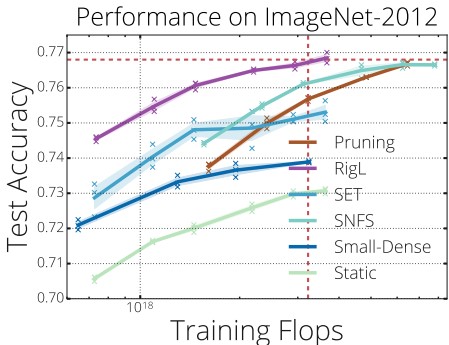 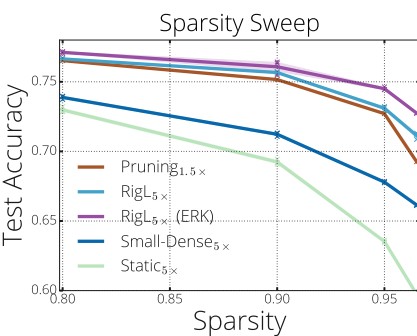

Figure 2: **(left)** Performance of various dynamic sparse training methods on ImageNet-2012 classification task. We use 80% sparse ResNet-50 architecture with uniform sparsity distribution. Points at each curve correspond to the individual training runs with training multipliers from 1 to 5 (except pruning which is scaled between 0.5 and 2). We repeat training 3 times at every multiplier and report the mean accuracies. The number of FLOPs required to train a standard dense Resnet-50 along with its performance is indicated with a dashed red line. **(right)** Performance of *RigL* at different sparsity levels with extended training. Results are averaged over 3 runs.

As long as $\Delta T > \frac{1}{1-s}$ the total work in calculating dense gradients is amortized and still proportional to $1 - S$. This is in contrast to the method of Dettmers & Zettlemoyer (2019), which requires calculating and storing the full gradients at each optimization step.

## 4 EMPIRICAL EVALUATION

Our experiments include image classification using CNNs on the ImageNet-2012 (Russakovsky et al., 2015) and CIFAR-10 (Krizhevsky et al.) datasets and character based language modelling using RNNs with the WikiText-103 dataset (Merity et al., 2016). We use the TensorFlow Model Pruning library (Zhu & Gupta, 2018) for our pruning baselines. A Tensorflow (Abadi et al., 2015) implementation of our method along with three other baselines (SET, SNFS, SNIP) will be open sourced. When we increase the training steps by a factor $M$, the anchor epochs of the learning rate schedule and the end iteration of the mask update schedule are also scaled by the same factor; we indicate this scaling with a subscript (e.g. $\text{RigL}_{M\times}$).

### 4.1 IMAGENET-2012 DATASET

In all experiments in this section, we use SGD with momentum as our optimizer. We set the momentum coefficient of the optimizer to 0.9, $L_2$ regularization coefficient to 0.0001, and label smoothing (Szegedy et al., 2016) to 0.1. The learning rate schedule starts with a linear warm up reaching its maximum value of 1.6 at epoch 5 which is then dropped by a factor of 10 at epochs 30, 70 and 90. We train our networks with a batch size of 4096 for 32000 steps which roughly corresponds to 100 epochs of training. Our training pipeline uses standard data augmentation, which includes random flips and crops.

### 4.1.1 RESNET-50

Figure 2-left summarizes the performance of various methods on training an 80% sparse ResNet-50. We also train small dense networks with equivalent parameter count. All sparse networks use the *constant* sparsity distribution and a cosine update schedule ($\alpha = 0.3$, $\Delta T = 100$). Overall, we observe that the performance of all methods improves with training time; thus, for each method we run extended training with up to $5\times$ the training steps of the original.

As noted by Gale et al. (2019), Evci et al. (2019), Frankle et al. (2019), and Mostafa & Wang (2019), training a network with fixed sparsity from scratch (*Static*) leads to inferior performance. Training a small dense network with the same number of parameters gets better results than *Static*, but fails to match the performance of dynamic sparse models. Similarly *SET* improves the performance over *Small-Dense*, however saturates around 75% accuracy indicating the limits of growing new connec-

| Method | Top-1 Accuracy | FLOPs (Train) | FLOPs (Test) | Top-1 Accuracy | FLOPs (Train) | FLOPs (Test) |
|---|---|---|---|---|---|---|
| Dense | 76.8±0.09 | 1x (3.2e18) | 1x (8.2e9) | | | |
| | | S=0.8 | | | S=0.9 | |
| Static | 70.6±0.06 | 0.23x | 0.23x | 65.8±0.04 | 0.10x | 0.10x |
| SNIP | 72.0±0.10 | 0.23x | 0.23x | 67.2±0.12 | 0.10x | 0.10x |
| Small-Dense | 72.1±0.12 | 0.20x | 0.20x | 68.9±0.10 | 0.12x | 0.12x |
| SET | 72.9±0.39 | 0.23x | 0.23x | 69.6±0.23 | 0.10x | 0.10x |
| RigL | 74.6±0.06 | 0.23x | 0.23x | 72.0±0.05 | 0.10x | 0.10x |
| Small-Dense$_{5\times}$ | 73.9±0.07 | 1.01x | 0.20x | 71.3±0.10 | 0.60x | 0.12x |
| RigL$_{5\times}$ | **76.6±0.06** | 1.14x | 0.23x | **75.7±0.06** | 0.52x | 0.10x |
| Static (ERK) | 72.1±0.04 | 0.42x | 0.42x | 67.7±0.12 | 0.24x | 0.24x |
| DSR* | 73.3 | 0.40x | 0.40x | 71.6 | 0.30x | 0.30x |
| RigL (ERK) | 75.1±0.05 | 0.42x | 0.42x | 73.0±0.04 | 0.25x | 0.24x |
| RigL$_{5\times}$ (ERK) | **77.1±0.06** | 2.09x | 0.42x | **76.4±0.05** | 1.23x | 0.24x |
| SNFS* | 74.2 | n/a | n/a | 72.3 | n/a | n/a |
| SNFS (ERK) | 75.2±0.11 | 0.61x | 0.42x | 72.9±0.06 | 0.50x | 0.24x |
| Pruning* (Zhu) | 73.2 | 1.00x | 0.23x | 70.3 | 1.00x | 0.10x |
| Pruning* (Gale) | 75.6 | 1.00x | 0.23x | 73.9 | 1.00x | 0.10x |
| Pruning$_{1.5\times}$ (Gale) | **76.5** | 1.50x | 0.23x | **75.2** | 1.50x | 0.10x |

Table 2: Performance and cost of sparse training methods on training 80% and 90% sparse ResNet-50s. FLOPs needed for training and test are normalized with the FLOPs of a dense model (see Appendix G for details on how FLOPs are calculated). Methods with a subscript indicate a rescaled training time, whereas '*' indicates reported results. (ERK) corresponds to the sparse networks with Erdős-Renyi-Kernel sparsity distribution. RigL$_{5\times}$ (ERK) achieves 77.1% Top-1 Accuracy using only 20% of the parameters of a dense model and 42% of its FLOPs.

tions randomly. Methods that use gradient information to grow new connections (*RigL* and *SNFS*) obtain higher accuracies, but *RigL* achieves the highest accuracy and does so while consistently requiring fewer FLOPs than the other methods.

Given that different applications or scenarios might require a limit on the number of FLOPs for inference, we investigate the performance of our method at various sparsity levels. As mentioned previously, one strength of our method is that its resource requirements are constant throughout training and we can choose the level of sparsity that fits our training and/or inference constraints. In Figure 2-right we show the performance of our method at different sparsities and compare them with the pruning results of Gale et al. (2019), which uses 1.5x training steps, relative to the original 32k iterations. To make a fair comparison with regards to FLOPs, we scale the learning schedule of all other methods by 5x. Note that even after extending the training, it takes less FLOPs to train sparse networks using *rigL* (except for the 80% sparse *RigL*-ERK) compared to the pruning method.

*RigL*, our method with constant sparsity distribution, **exceeds** the performance of magnitude based iterative pruning in all sparsity levels while requiring less FLOPs to train. Sparse networks that use *Erdős-Renyi-Kernel (ERK)* sparsity distribution obtains even greater performance. For example ResNet-50 with 96.5% sparsity achieves a remarkable 72.75% Top-1 Accuracy, around 3.5% higher than the extended magnitude pruning results reported by Gale et al. (2019). As observed earlier, smaller dense models (with the same number of parameters) or sparse models with a static connectivity can not perform at a comparable level.

A more fine grained comparison of sparse training methods is presented in Table 2. Methods using uniform sparsity distribution and whose FLOP/memory footprint scales directly with (1-S) are placed in the first sub-group of the table. The second sub-group includes DSR and networks with ERK sparsity distribution which require a higher number of FLOPs for inference with same parameter count. The final sub-group includes methods that require the space and the work proportional to training a dense model.

---

*approximated

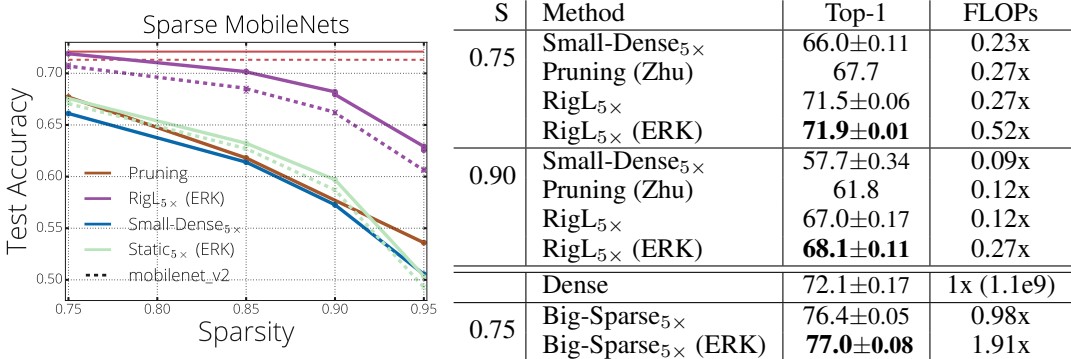

| S | Method | Top-1 | FLOPs |
|---|---|---|---|
| 0.75 | Small-Dense$_{5\times}$ | 66.0±0.11 | 0.23x |
| | Pruning (Zhu) | 67.7 | 0.27x |
| | RigL$_{5\times}$ | 71.5±0.06 | 0.27x |
| | RigL$_{5\times}$ (ERK) | **71.9±0.01** | 0.52x |
| 0.90 | Small-Dense$_{5\times}$ | 57.7±0.34 | 0.09x |
| | Pruning (Zhu) | 61.8 | 0.12x |
| | RigL$_{5\times}$ | 67.0±0.17 | 0.12x |
| | RigL$_{5\times}$ (ERK) | **68.1±0.11** | 0.27x |
| | Dense | 72.1±0.17 | 1x (1.1e9) |
| 0.75 | Big-Sparse$_{5\times}$ | 76.4±0.05 | 0.98x |
| | Big-Sparse$_{5\times}$ (ERK) | **77.0±0.08** | 1.91x |

Figure 3: **(left)** *RigL* significantly improves the performance of Sparse MobileNets on ImageNet-2012 dataset and exceeds the *pruning* results reported by Zhu & Gupta (2018). Performance of the dense MobileNets are indicated with red lines. **(right)** Performance of sparse MobileNet-v1 architectures presented with their inference FLOPs. Networks with *ERK* distribution get better performance with the same number of parameters but take more FLOPs to run. Training wider sparse models with *RigL* (*Big-Sparse*) yields a significant performance improvement over the dense model.

### 4.1.2 MOBILENET

MobileNet is a compact architecture that performs remarkably well in resource constrained settings. Due to its compact nature with separable convolutions it is known to be difficult to sparsify (Zhu & Gupta, 2018). In this section we apply our method to MobileNet-v1 (Howard et al., 2017) and MobileNet-v2 (Sandler et al., 2018). Due to its low parameter count we keep the first layer dense, and use ERK and Uniform sparsity distributions to sparsify the remaining layers.

The performance of sparse MobileNets trained with *RigL* as well as the baselines are shown in Figure 3. We do extended training (5x of the original number of steps) for all runs in this section. Although MobileNets are more sensitive to sparsity compared to the ResNet-50 architecture, *RigL* successfully trains sparse MobileNets at high sparsities and exceeds the performance of previously reported pruning results.

To demonstrate the advantages of sparse models, next, we train wider MobileNets while keeping the FLOPs and total number of parameters the same as the dense baseline using sparsity. A sparse MobileNet-v1 with width multiplier 1.98 and constant 75% sparsity has the same FLOPs and parameter count as the dense baseline. Training this network with *RigL* yields an impressive **4.3% absolute improvement** in Top-1 Accuracy.

### 4.2 CHARACTER LEVEL LANGUAGE MODELLING

Most prior work has only examined sparse training on vision networks [the exception is the earliest work - Deep Rewiring (Bellec et al., 2017) which trained an LSTM (Hochreiter & Schmidhuber, 1997) on the TIMIT (Garofolo et al., 1993) dataset]. To fully understand these techniques it is important to examine different architectures on different datasets. Kalchbrenner et al. (2018) found sparse GRUs (Cho et al., 2014) to be very effective at modeling speech, however the dataset they used is not available. We choose a proxy task with similar characteristics (dataset size and vocabulary size are approximately the same) - character level language modeling on the publicly available WikiText-103 (Merity et al., 2016) dataset.

Our network consists of a shared embedding with dimensionality 128, a vocabulary size of 256, a GRU with a state size of 512, a readout from the GRU state consisting of two linear layers with 256 units and 128 units respectively. We train the next step prediction task with the standard cross entropy loss, the Adam optimizer, a learning rate of $7e-4$, an L2 regularization coefficient of $5e-4$, a sequence length of 512, a batch size of 32 and gradient absolute value clipping of values larger (in magnitude) than 10. Baseline training length is 200,000 iterations. When inducing sparsity with magnitude pruning (Zhu & Gupta, 2018), we perform pruning between iterations 50,000 and

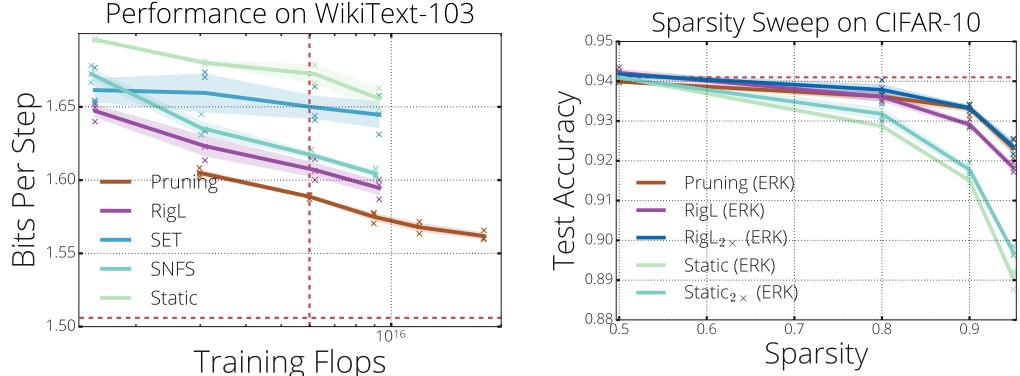

Figure 4: **(left)** Final validation loss of various sparse training methods on character level language modelling task. Cross entropy loss is converted to bits (from nats). Performance and the training cost of a dense model is indicated with dashed red lines. **(right)** Test accuracies of sparse WideResNet-22-2's on CIFAR-10 task.

150,000 with a pruning frequency of 1,000. We initialize sparse networks with a uniform sparsity distribution and use a cosine update schedule with $\alpha = 0.1$ and $\Delta T = 100$. Unlike the previous experiments we keep updating the mask until the end of the training; we observed this performed slightly better than stopping at iteration 150,000.

In Figure 4-left we report the validation loss of various solutions at the end of the training. For each method we perform extended runs to see how they scale with increasing training time. As observed before, SET performs worst than the other dynamic training methods and its performance improves only slightly with increased training time. On the other hand the performance of *RigL* and SNFS improves constantly with more training steps. Both of these methods falls short of matching the pruning performance.

### 4.3 WIDERESNET-22-2 ON CIFAR-10

In this section, we evaluate the performance of *RigL* on CIFAR-10 image classification benchmark. We train Wide Residual Network's (Zagoruyko & Komodakis, 2016) with 22 layers using a width multiplier of 2 for 250 epochs (97656 steps). Learning rate starts at 0.1 and scaled down by a factor of 5 every 30,000 iterations. We use an L2 regularization coefficient of 5e-4, a batch size of 128 and a momentum coefficient of 0.9. We keep the hyper-parameters specific to *RigL* same as the ImageNet experiments, except the final iteration for mask updates; which is adjusted to 75000. Results with different mask update intervals can be found in Appendix H.

Performance of *RigL* across different sparsity levels is presented in Figure 4-right. Corresponding final training losses of the trained networks can be found in Appendix H. The dense baseline obtains 94.1% test accuracy. Networks with half of the connections removed (50% sparsity) achieves roughly the same accuracy as the dense baseline. Surprisingly, some of the networks at this sparsity level generalize better than the dense baseline demonstrating the regularization aspect of using sparsity. With increased sparsity, we start to see a performance gap between the *Static* and *Pruning* solutions. Training static *RigL* networks longer seems to have limited effect on the final performance. On the other hand, *RigL*, matches the performance of pruning using only a fraction of resources needed for training a dense network.

### 4.4 ANALYZING THE PERFORMANCE OF *RigL*

In this section we study the effect of *sparsity distributions*, *update schedules*, and *dynamic connections* on the performance of our method. The results for SET and SNFS are similar and are discussed in Appendices B and E.

**Effect of Mask Initialization:** Figure 5-left shows how the sparsity distribution affects the final test accuracy of sparse ResNet-50s trained with *RigL*. Erdős-Rényi-Kernel (ERK) performs consistently better than the other two distributions. ERK automatically allocates more parameters to the

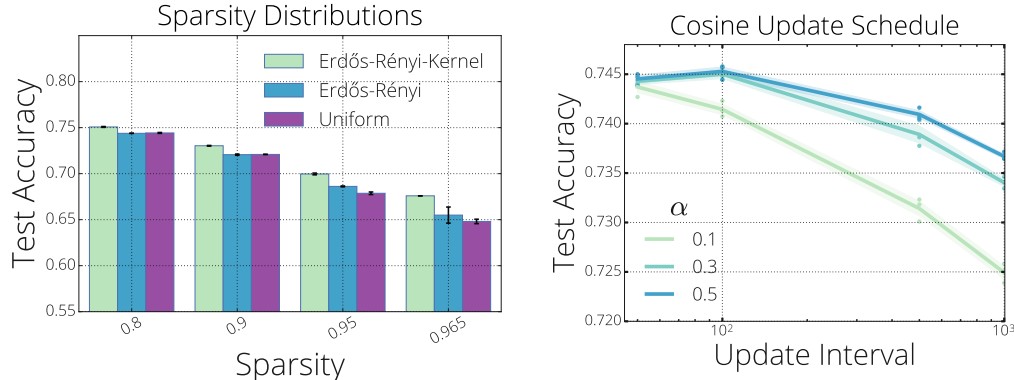

Figure 5: **(left)** Performance of *RigL* at different sparsities using different sparsity masks **(right)** Ablation study on cosine schedule. Other methods are in the appendix.

layers with few parameters by decreasing their sparsities[†]. This reallocation seems to be crucial for preserving the capacity of the network at high sparsity levels where ERK outperforms other distributions by a greater margin. Though it performs better, the ERK distribution requires approximately twice as many FLOPs compared to a uniform distribution. This highlights an interesting trade-off between accuracy and computational efficiency even though both models have the same number of parameters.

**Effect of Update Schedule and Frequency:** In Figure 5-right, we evaluate the performance of our method on update intervals $\Delta T \in [50, 100, 500, 1000]$ and initial drop fractions $\alpha \in [0.1, 0.3, 0.5]$. The best accuracies are obtained when the mask is updated every 100 iterations with an initial drop fraction of 0.3 or 0.5. Notably, even with frequent update intervals (e.g. every 1000 iterations), *RigL* performs above 73.5%.

**Effect of Dynamic connections:** Frankle et al. (2019) and Mostafa & Wang (2019) observed that static sparse training converges to a solution with a higher loss than dynamic sparse training. In Figure 6-left we examine the loss landscape lying between a solution found via static sparse training and a solution found via pruning to understand whether former lies in a basin isolated from the latter. Performing a linear interpolation between the two reveals the expected result – high-loss barrier – demonstrating that the loss landscape is not trivially connected. However, this is only one of infinitely many paths between the two points optimization can be used to find parametric curves that connects solutions (Garipov et al., 2018; Draxler et al., 2018) subject to constraints. For example Garipov et al. (2018) showed different dense solutions lie in the same basin by finding 2nd order Bézier curves with low energy between the two solutions. Following their method, we attempt

---

[†]see Appendix I for exact layer-wise sparsities given by ERK for 80% and 90% sparse ResNet-50's.

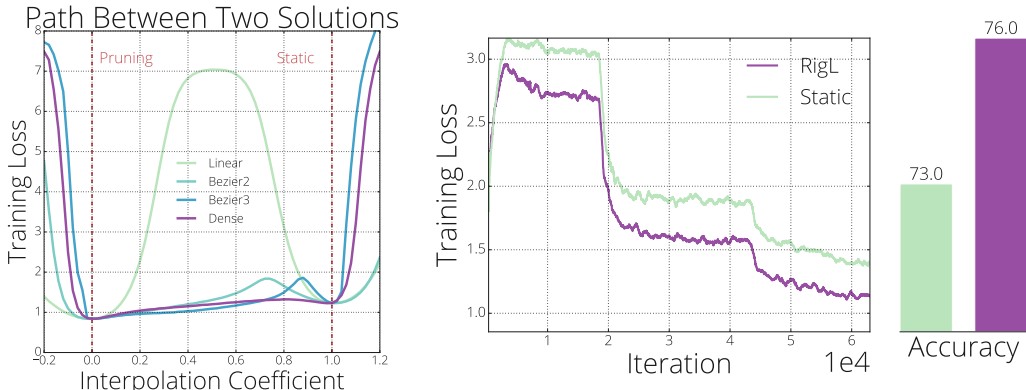

Figure 6: **(left)** Training loss evaluated at various points on interpolation curves between a magnitude pruning model (0.0) and a model trained with static sparsity (1.0). **(right)** Training loss of *RigL* and *Static* methods starting from the static sparse solution, and their final accuracies.

to find quadratic and cubic Bézier curves between the two sparse solutions. Surprisingly, even with a cubic curve, we fail to find a path without a high-loss barrier. These results suggest that static sparse training can get stuck at local minima that are isolated from improved solutions. On the other hand, when we optimize the quadratic Bézier curve across the full **dense** space we find a near-monotonic path to the improved solution, suggesting that allowing new connections to grow lends dynamic sparse training greater flexibility in navigating the loss landscape.

In Figure 6-right we train *RigL* starting from the sub-optimal solution found by static sparse training, demonstrating that it is able to escape the local minimum, whereas re-training with static sparse training cannot. *RigL* first removes connections with the smallest magnitudes since removing these connections have been shown to have a minimal effect on the loss (Han et al., 2015; Evci, 2018). Next, it activates connections with the high gradients, since these connections are expected to decrease the loss fastest. We hypothesize in Appendix A that *RigL* escapes bad critical points by replacing saddle directions with high gradient dimensions.

## 5 DISCUSSION & CONCLUSION

In this work we introduced 'Rigged Lottery' or *RigL*, an algorithm for training sparse neural networks efficiently. For a given computational budget *RigL* achieves higher accuracies than existing dense-to-sparse and sparse-to-sparse training algorithms. *RigL* is useful in three different scenarios: (1) To improve the accuracy of sparse models intended for deployment; (2) To improve the accuracy of large sparse models which can only be trained for a limited number of iterations; and (3) Combined with sparse primitives to enable training of extremely large sparse models which otherwise would not be possible.

The third scenario is unexplored due to the lack of hardware and software support for sparsity. Nonetheless, work continues to improve the performance of sparse networks on current hardware (Hong et al., 2019; Merrill & Garland, 2016), and new types of hardware accelerators will have better support for parameter sparsity (Wang et al., 2018; Mike Ashby, 2019; Liu et al., 2018; Han et al., 2016a; Chen et al., 2019). *RigL* provides the tools to take advantage of, and motivation for, such advances.

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

## A  EFFECT OF MASK UPDATES ON THE ENERGY LANDSCAPE

To update the connectivity of our sparse network, we first need to drop a fraction $d$ of the existing connections for each layer independently to create a budget for growing new connections. Following the recipe of magnitude based pruning(Han et al., 2015), we order parameters at layer $i$ by magnitude $|\theta_i|$ and drop the $N * (1 - S) * d$ parameters with lowest magnitude. The effectiveness of this simple criteria can be explained through the first order Taylor approximation of the loss $L$ around the current set of parameters $\theta$.

$$\Delta L = L(\theta + \Delta\theta) - L(\theta) = \nabla_\theta L(\theta)\Delta\theta + R(||\Delta\theta||_2^2)$$

The main goal of dropping connections is to remove parameters with minimal impact on the neural network and therefore on its performance. Since removing the connection $\theta_i$ corresponds to setting it to zero, it incurs a change of $\Delta\theta = -\theta_i$ in that direction and a change of $\Delta L_i = -\nabla_{\theta_i} L(\theta)\theta_i + R(\theta_i^2)$ in the loss, where the first term is usually defined as the *saliency* of a connection. Though using saliency to remove connections has been used as a criteria for removing connections (Molchanov et al., 2016), it has been shown to produce inferior results compared to magnitude

based removal, especially when used to remove multiple connections at once (Evci, 2018). In contrast, picking the lowest magnitude connections ensures a small remainder term in addition to a low saliency, limiting the damage we make when we drop connections. Additionally, we note that connections with small magnitude can only remain small if the gradient is also small, meaning that the saliency is likely small when the parameter itself is small. Therefore we argue that the connections removed by *RigL* are likely to be saddle directions of the energy landscape.

After the removal of insignificant connections, we enable new connections that have the highest expected gradients. Since we initialize these new connections to zero, they are guaranteed to have high gradients in the proceeding iteration and therefore to reduce the loss quickly. By definition a direction with high magnitude gradient is not a saddle direction. Combining this observation with the previous (*RigL* is likely to remove saddle directions) and the results in Section 4.4 we suggest that *RigL* improves the energy landscape of the optimization by replacing saddle directions with the ones with high gradient. This helps the optimization procedure to escape bad critical points and find solutions with higher quality.

## B    EFFECT OF SPARSITY DISTRIBUTION ON OTHER METHODS

In Figure 7-left we show the effect of sparsity distribution choice on 4 different sparse training methods. ERK distribution performs better than other distributions for each training method.

## C    EFFECT OF MOMENTUM COEFFICIENT FOR SNFS

In Figure 7 right we show the effect of the momentum coefficient on the performance of SNFS. Our results shows that using a coefficient of 0.99 brings the best performance. On the other hand using the most recent gradient only (coefficient of 0) performs as good as using a coefficient of 0.9. This result might be due to the large batch size we are using (4096), but it still motivates using *RigL* and instantaneous gradient information only when needed, instead of accumulating them.

## D    EXISTENCE OF LOTTERY TICKETS

We perform the following experiment to see whether *Lottery Tickets* exist in our setting. We take the sparse network found by *RigL* and restart training using original initialization, both with RigL and with fixed topology as in the original Lottery Ticket Hypothesis. Results in table 3 demonstrate that training with a fixed topology is significantly worse than training with *RigL* and that *RigL* does not benefit from starting again with the final topology and the original initialization - training for twice as long instead of rewiring is more effective. In short, there are no special tickets, with *RigL* all tickets seems to win.

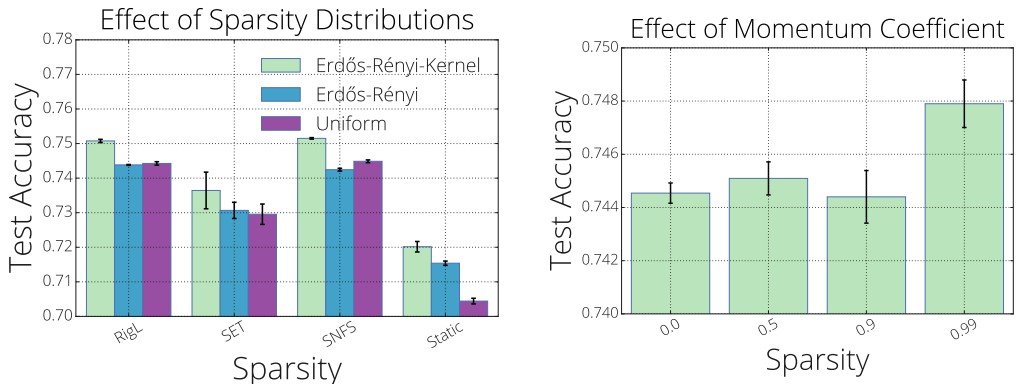

Figure 7: **(left)** Effect of sparsity distribution choice on sparse training methods at different sparsity levels. We average over 3 runs and report the standard deviations for each. **(right)** Effect of momentum value on the performance of SNFS algorithm. Setting the momentum coefficient of the SNFS algorithm to 0 seems to perform best, suggesting the accumulated values are not important.

| Initialization | Training Method | Test Accuracy | Training FLOPs |
|----------------|-----------------|---------------|----------------|
| Lottery | Static | 70.82±0.07 | 0.46x |
| Lottery | RigL | 73.93±0.09 | 0.46x |
| Random | RigL | 74.55±0.06 | 0.23x |
| Random | RigL$_{2\times}$ | 76.06±0.09 | 0.46x |

Table 3: Effect of lottery ticket initialization on the final performance. There are no special tickets and dynamic connectivity provided by *RigL* is critical for good performance.

## E    EFFECT OF UPDATE SCHEDULES ON OTHER DYNAMIC SPARSE METHODS

In Figure 8 we repeat the hyper-parameter sweep done for *RigL* in Figure 5-right, using SET and SNFS. Cosine schedule with $\Delta T = 50$ and $\alpha = 0.1$ seems to work best across all methods. An interesting observation is that higher drop fractions ($\alpha$) seem to work better with longer intervals $\Delta T$. For example, SET with $\Delta T = 1000$ seems to work best with $\alpha = 0.5$.

## F    ALTERNATIVE UPDATE SCHEDULES

In Figure 9, we share the performance of two alternative annealing functions:

1. *Constant:* $f_{decay}(t) = \alpha$.
2. *Inverse Power:* The fraction of weights updated decreases similarly to the schedule used in Zhu & Gupta (2018) for iterative pruning: $f_{decay}(t) = \alpha(1 - \frac{t}{T_{end}})^k$. In our experiments we tried $k = 1$ which is the linear decay and their default $k = 3$.

*Constant* seems to perform well with low initial drop fractions like $\alpha = 0.1$, but it starts to perform worse with increasing $\alpha$. *Inverse Power* for k=3 and k=1 (*Linear*) seems to perform similarly for low $\alpha$ values. However the performance drops noticeably for k=3 when we increase the update interval. As reported by Dettmers & Zettlemoyer (2019) linear (k=1) seems to provide similar results as the cosine schedule.

## G    CALCULATING FLOPS OF MODELS AND METHODS

In order to calculate FLOPs needed for a single forward pass of a sparse model, we count the total number of multiplications and additions layer by layer for a given layer sparsity $s^l$. The total FLOPs is then obtained by summing up all of these multiply and adds.

Different sparsity distributions require different number of FLOPs to compute a single prediction. For example *Erdős-Renyi-Kernel* distributions usually cause earlier layers to be less sparse than the later layers (see Appendix I). The inputs of earlier layers have greater spatial dimensions, so

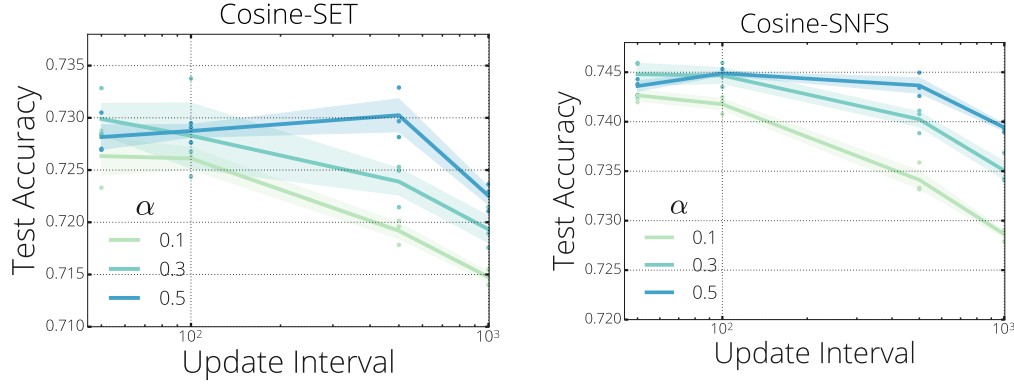

Figure 8: Cosine update schedule hyper-parameter sweep done using dynamic sparse training methods SET **(left)** and SNFS **(right)**.

a convolutional kernel that works on such inputs will require more FLOPs to compute the output features compared to later layers. Thus, having earlier layers which are less sparse results in a higher total number of FLOPs required by a model.

Training a neural network consists of 2 main steps:

1. *forward pass:* Calculating the loss of the current set of parameters on a given batch of data. During this process layer activations are calculated in sequence using the previous activations and the parameters of the layer. Activation of layers are stored in memory for the backward pass.

2. *backward pass:* Using the loss value as the initial error signal, we back-propagate the error signal while calculating the gradient of parameters. During the backward pass each layer calculates 2 quantities: the gradient of the activations of the previous layer and the gradient of its parameters. Therefore in our calculations we count backward passes as two times the computational expense of the forward pass. We omit the FLOPs needed for batch normalization and cross entropy.

Dynamic sparse training methods require some extra FLOPs to update the connectivity of the neural network. We omit FLOPs needed for dropping the lowest magnitude connections in our calculations. For a given dense architecture with FLOPs $f_D$ and a sparse version with FLOPs $f_S$, the total FLOPs required to calculate the gradient on a single sample is computed as follows:

- **Static Sparse and Dense.** Scales with $3 * f_S$ and $3 * f_D$ FLOPs, respectively.

- **Snip.** We omit the initial dense gradient calculation since it is negligible, which means Snip scales in the same way as Static methods: $3 * f_S$ FLOPs.

- **SET.** We omit the extra FLOPs needed for growing random connections, since this operation can be done on chip efficiently. Therefore, the total FLOPs for SET scales with $3 * f_S$.

- **SNFS.** Forward pass and back-propagating the error signal needs $2 * f_S$ FLOPs. However, the dense gradient needs to be calculated at every iteration. Thus, the total number of FLOPs scales with $2 * f_S + f_D$.

- **RigL.** Iterations with no connection updates need $3 * f_S$ FLOPs. However, at every $\Delta T$ iteration we need to calculate the dense gradients. This results in the average FLOPs for *RigL* given by $\frac{(3*f_S*\Delta T+2*f_S+f_D)}{(\Delta T+1)}$.

# H    ADDITIONAL PLOTS AND EXPERIMENTS FOR CIFAR-10

In Figure 10-left, we plot the final training loss of experiments presented in Section 4.3 to investigate the generalization properties of the algorithms considered. Poor performance of *Static* reflects itself in training loss clearly across all sparsity levels. *RigL* achieves similar final loss as the pruning, despite having around half percent less accuracy. Training longer with *RigL* decreases the final loss further and the test accuracies start matching pruning (see Figure 4-right) performance. These results show that *RigL* improves the optimization as promised, however generalizes slightly worse than pruning.

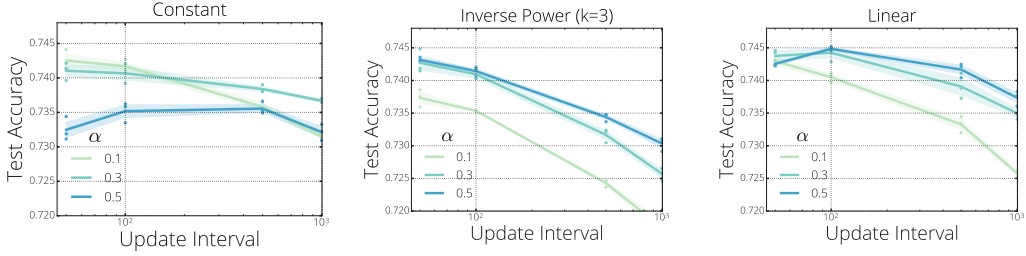

Figure 9: Using other update schedules with *RigL*: **(left)** Constant **(middle)** Exponential (k=3) and **(right)** Linear

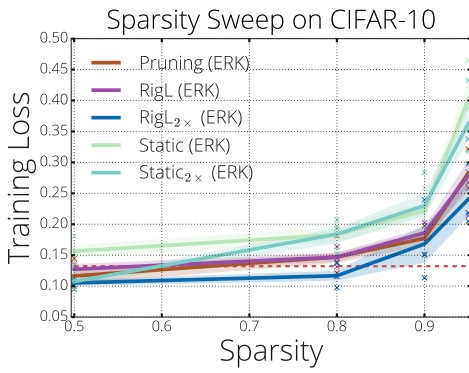 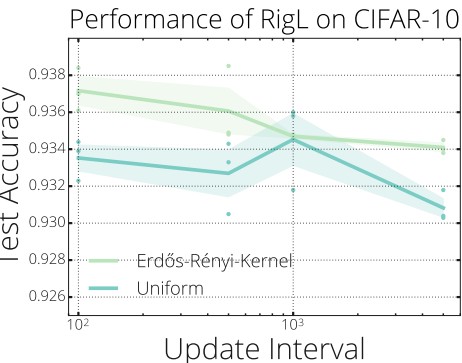

Figure 10: Final training loss of sparse models **(left)** and performance of *RigL* at different mask update intervals **(right)**.

In Figure 10-right, we sweep mask update interval $\Delta T$ and plot the final test accuracies. We fix initial drop fraction $\alpha$ to 0.3 and evaluate two different sparsity distributions: *Uniform* and *ERK*. Both curves follow a similar pattern as in Imagenet-2012 sweeps (see Figure 8) and best results are obtained when $\Delta T = 100$.

## I SPARSITY OF INDIVIDUAL LAYERS FOR SPARSE RESNET-50

Sparsity of ResNet-50 layers given by the Erdős-Rényi-Kernel sparsity distribution plotted in Figure 11.

## J BUGS DISCOVERED DURING EXPERIMENTS

Our initial implementations contained some subtle bugs, which while not affecting the general conclusion that *RigL* is more effective than other techniques, did result in lower accuracy for all sparse training techniques. We detail these issues here with the hope that others may learn from our mistakes.

1. **Random operations on multiple replicas.** We use data parallelism to split a mini-batch among multiple replicas. Each replica independently calculates the gradients using a different sub-mini-batch of data. The gradients are aggregated using an ALL-REDUCE operation before the optimizer update. Our implementation of SET, SNFS and *RigL* depended on each replica independently choosing to drop and grow the same connections. However, due to the nature of random operations in Tensorflow, this did not happen. Instead, different replicas diverged after the first drop/grow step. This was most pronounced in SET where each replica chose at random and much less so for SNFS and *RigL* where randomness is only needed to break ties. If left unchecked this might be expected to be catastrophic, but due to the behavior of Estimators and/or TF-replicator, the values on the first replica are broadcast to the others periodically (every approximately 1000 steps in our case).

   We fixed this bug by using stateless random operations. As a result the performance of SET improved slightly (0.1-0.3 % higher on Table 2).

2. **Synchronization between replicas.** *RigL* and SNFS depend on calculating dense gradients with respect to the masked parameters. However, as explained above, in the multiple replica setting these gradients need to be aggregated. Normally this aggregation is automatically done by the optimizer, but in our case, this does not happen (only the gradients with respect to the *unmasked* parameters are aggregated automatically). This bug affected SNFS and *RigL*, but not SET since SET does not rely on the gradients to grow connections. Again, the synchronization of the parameters from the first replica every approximately 1000 steps masked this bug.

   We fixed this bug by explicitly calling ALL-REDUCE on the gradients with respect to the masked parameters. With this fix, the performance of *RigL* and SNFS improved significantly, particularly for default training lengths (around 0.5-1% improvement).

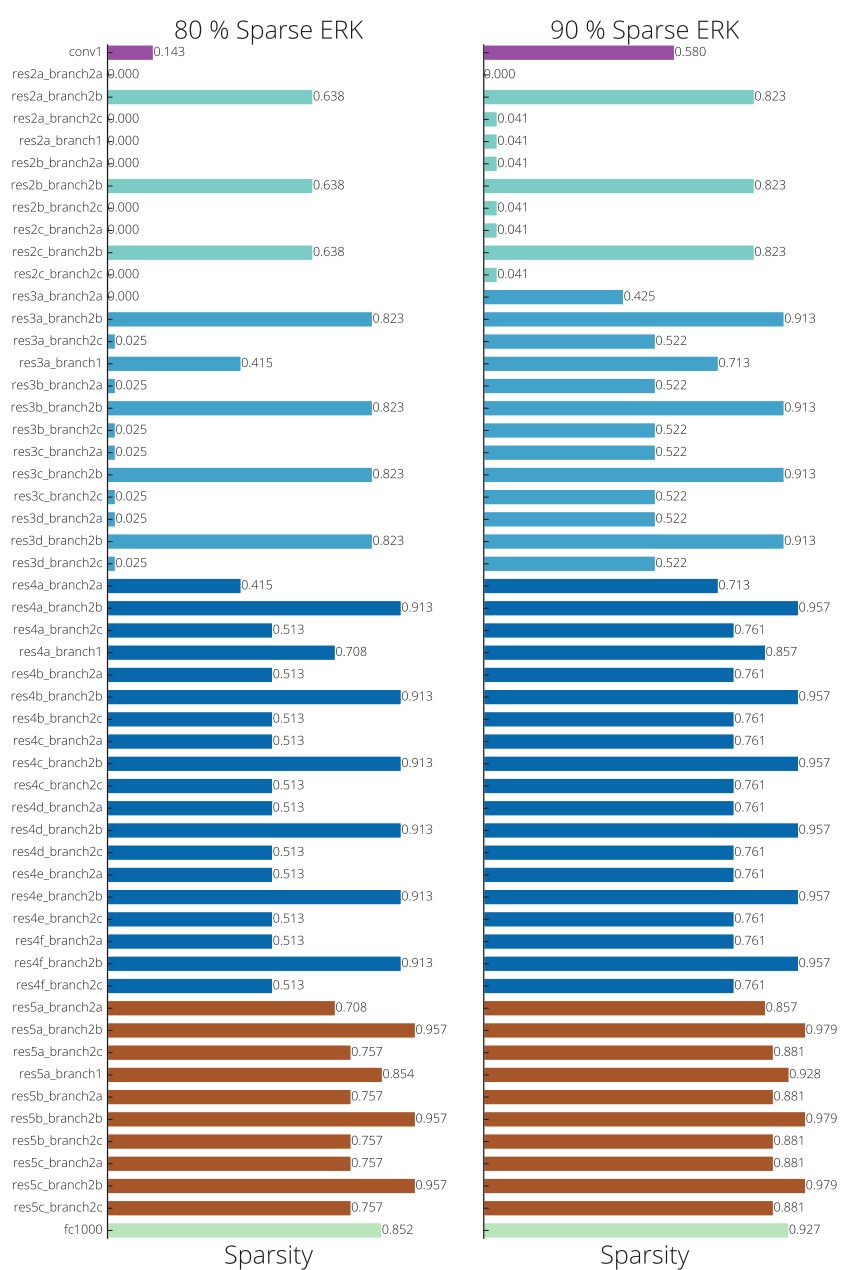

Figure 11: Sparsities of individual layers of the ResNet-50.

3. **SNIP Experiments.** Our first implementation of SNIP used the gradient magnitudes to decide which connections to keep causing its performance to be worse than static. Upon our discussions with the authors of SNIP, we realized that the correct metric is the saliency (gradient times parameter magnitude). With this correction SNIP performance improved dramatically to better than random (Static) even at Resnet-50/ImageNet scale. It is surprising that picking connections with the highest gradient magnitudes can be so detrimental to training (it resulted in much worse than random performance).

