# OpenReview forum: "Rigging the Lottery: Making All Tickets Winners"
_ICLR.cc/2020/Conference — Reject_

### Official Review · AnonReviewer3 · 2019-10-22
**Official Blind Review #3**

**Rating:** 6

**Review:**

Overview:

The paper is dedicated to developing a more efficient and powerful dense-to-sparse training method. In order to break the limits of the size of the largest trainable sparse model to that of the largest trainable dense model, the author proposes a dynamic method that updates the network topology via parameter magnitudes and infrequent gradient calculation. In the experiments parts, they conduct extensive studies to show the proposed approach can surpass the previous sota with ResNet-50, MobileNet v1 and v2 on the imagenet 2012. What's more, the author also provides some intuitive explanation about why allowing topology change during the optimization is beneficial.

Strength Bullets:

1. Due to the dynamic network topology, the paper's methods exactly achieve the memory and computation efficient. i) Required memory is only proportional to the size of the sparse model. ii) The amount of computation is proportional to the number of nonzero parameters in the model.
2. The author performs detailed comparison experiments among different sparsity distribution and different pruning methods. And the results overcome the previous state-of-the-art results.
3. Fig 5 shows some interesting insight. It suggests that static sparse training may be stuck at some local minima which are isolated from improved solutions. However, the dynamic update has a big chance to avoid this problem.

Weakness Bullets:

1. The author claims that the ticket in the paper does not rely on a "lucky" initialization. But it doesn't exclude the possibility that starting from the original initial conditions may give a better performance. Even if the connection is dynamic, we can still record the initial point for each weight. It will be better the author can provide related analysis.
2. In my opinion, in order to prove dynamic pruning is better than static methods, the author needs to provide a comparison with the previous sota method in it's own setting. i.e. The Lottery Ticket Hypothesis: Finding Sparse, Trainable Neural Networks

Recommendation:

I think this paper is a novel work. Although it has some flaws in the experiment design, the motivation and experiment results are conniving enough. So, this is a weak accept.

**Experience Assessment:**

I have read many papers in this area.

**Review Assessment: Checking Correctness Of Derivations And Theory:**

I carefully checked the derivations and theory.

**Review Assessment: Checking Correctness Of Experiments:**

I assessed the sensibility of the experiments.

**Review Assessment: Thoroughness In Paper Reading:**

I read the paper at least twice and used my best judgement in assessing the paper.

---

> ### Author Response · Authors · 2019-11-12
> **Response from the Authors**
>
> We appreciate Reviewer 3's comments and address the weakness bullets below:
>
> (1) We add experiments that demonstrate that choosing the lottery initialization in this setting does not improve the results (Appendix D). There are no special tickets. Rewiring RigL solution to the initial conditions gets 70.82 % average accuracy, which is only 0.2% better than `Static` training. Results demonstrate that training with a fixed topology is significantly worse than training with RigL and that RigL does not benefit from starting again with the final topology and original initialization - training for twice as long is more effective than restarting the training.
>
> (2) We aren’t quite sure what experiments you want to see when you say compare to Lottery “in its original setting”.  If we included lottery ticket experiments, the cost of doing Iterative Magnitude Pruning (5 times for 80% sparse and 10 times for 90% sparse) would put the cost far off the scale of Figure 2 (approximately all the way to the edge of the page). We also like to point out that, for all of our experiments , we include pruning results which are approximately an upper bound on the performance of lottery tickets at much lower computational cost.

---

### Official Review · AnonReviewer2 · 2019-10-22
**Official Blind Review #2**

**Rating:** 6

**Review:**

This paper proposes a method for training sparse network without first training a dense network (e.g. the lottery ticket hypothesis or distillation). The method involves a combination of dynamic pruning of weights coupled with a dynamic "growing" of new weights given a novel criterion based on the magnitude of the gradient of the loss. As a result, networks can stay sparse throughout training and testing, leading to a large reduction in computational cost.

The paper's approach of dynamically changing the topology of networks is an interesting and motivated idea that seems to work rather well. I also appreciate the experiments on MobileNet, a setting where one expects investigations into sparse network architectures to have significant application. Relatedly, I appreciate the importance of the fact that the computational cost of training and evaluating the network is proportional to the sparse model size, which is not normally true for masked dense models. Overall, I found the paper to be very clear and of high quality, and thus I find this to be an interest addition to investigations into the lottery ticket hypothesis.

As the authors state, the novelty of their method is that they use the gradients with the highest magnitudes to grow connections. This is somewhat intuitive given the role gradients play in gradient descent based optimization, but I was wondering if they had any further intuition as to why this is the right criterion?

In section 4.3, I was a bit confused by Figure 5. My understanding is that many paths between loss landscape minima follow nonlinear paths -- why is it at all significant that there's a linear barrier? Why are only quadric and cubic Bezier curves used, rather than a more general path finding algorithm?

Overall, this is a nice paper that should be accepted to ICLR.

**Experience Assessment:**

I have read many papers in this area.

**Review Assessment: Checking Correctness Of Derivations And Theory:**

I assessed the sensibility of the derivations and theory.

**Review Assessment: Checking Correctness Of Experiments:**

I assessed the sensibility of the experiments.

**Review Assessment: Thoroughness In Paper Reading:**

I read the paper at least twice and used my best judgement in assessing the paper.

---

> ### Author Response · Authors · 2019-11-12
> **Response from the Authors**
>
> We would like to thank Reviewer 2 for their time and address the points raised:
>
> (1) Appendix F (now Appendix A) “Effect of Mask Updates on the Energy Landscape” contains an explanation based on a Taylor series approximation of why magnitude pruning and largest gradient growth is optimal (at least greedily). Unfortunately we did not originally reference this in the main text.  We have updated the last paragraph of Section 4 with a short discussion and a pointer as the following:
> "...RigL first removes connections with the smallest magnitudes since removing these connections have been shown to have a minimal effect on the loss. Next, it activates connections with the high gradients, since these connections are expected to decrease the loss fastest. We hypothesize in Appendix A that RigL escapes bad critical points by replacing saddle directions with high gradient dimensions."
>
> (2) The linear path having a high loss is only to demonstrate that the connectivity is not simple - it is the expected result.  We note that prior work (https://arxiv.org/pdf/1802.10026.pdf) was able to use a quadratic Bezier curve to connect minima in dense networks, so it seemed a reasonable choice.  Given N anchor points, the Bezier curves are capable of representing a more general path than the other technique we are aware of for connecting minima (http://proceedings.mlr.press/v80/draxler18a/draxler18a.pdf).  In both cases we must solve an optimization problem involving all N anchor points simultaneously.  We run out of memory when attempting to use a 4th order curve.

---

### Official Review · AnonReviewer1 · 2019-10-26
**Official Blind Review #1**

**Rating:** 3

**Review:**

This paper proposes a pruning technique called RigL, which performs sparse initialization of the network weights, and allows the network to grow weights during training. The sparse initialization makes the pruning algorithm to be memory- and computation- efficient, unlike existing models that starts with dense network weights. The authors train RigL with three different sparsity distributions that considers the number of input and output nodes, and validate them on various deep convnets for training on ImageNet, on which the model outperforms several existing sparsification methods and even dense counterparts.

Pros
- The proposed model, RigL, is memory- and computation- efficient, and thus allows to train a large network in an efficient manner.

- RigL obtains impressive sparsification performance, even yielding sparse networks that outperform their dense counterparts.

Cons
- The idea of starting from a small, sparse network and expanding it is not novel. DEN [Yoon et al. 18] proposed the same bottom-up approach with sparsely initialized networks, while they allowed to increase the number of neurons at each layer and focused more on continual learning. The authors should compare the two methods both conceptually and experimentally.

- The method is more like a set of heuristics rather than a principled approach, which makes it less appealing. This is not really an issue if the paper includes extensive experimental validation and in-depth analysis, but this is not the case.

- The experimental validation is largely lacking, as the authors only perform experiments on ImageNet and do not compare against recent state-of-the-art Bayesian sparsification methods (SBP, VIB, L0-regularization). Without such extensive experimental validation, it is uncertain whether the result will generalize, given the highly empirical nature of the work.

In sum, although I believe that the paper proposes a very practical method that is easy to implement and is promising, due to lack of experimental validation against a similar approach, state-of-the-art sparsification methods, and results on more datasets, I temporarily provide the rating of weak reject. I may change my opinion if the authors provide those results during the rebuttal period.

[Yoon et al. 18] Lifelong learning with dynamically expandable networks, ICLR 2018

**Experience Assessment:**

I have published in this field for several years.

**Review Assessment: Checking Correctness Of Derivations And Theory:**

I carefully checked the derivations and theory.

**Review Assessment: Checking Correctness Of Experiments:**

I carefully checked the experiments.

**Review Assessment: Thoroughness In Paper Reading:**

I read the paper thoroughly.

---

> ### Author Response · Authors · 2019-11-12
> **Response from the Authors**
>
> We would like to thank Reviewer 1 and respond to their concerns below:
>
> (1) Thank you for the reference to DEN, it is an interesting work. However, there are some important differences between our work and DEN:
>   (a) As emphasized in its abstract and in the introduction DEN attacks the problem of catastrophic forgetting in the continual learning setting and grows new neurons during training to efficiently learn each task in sequence. In contrast, we fix the number of available neurons in each layer and focus on sparse training of neural networks for a single task.  In theory we could compare the cost of training with these two different approaches, however:
>   (b) When extending continual learning (as used by DEN) to the split dataset regime it is non-trivial to match the performance of a standard training regime (as used by RigL).  For example, see http://openaccess.thecvf.com/content_CVPR_2019/papers/Wu_Large_Scale_Incremental_Learning_CVPR_2019_paper.pdf .  We are not aware of work that is able to do this successfully, and would appreciate references so that we can possibly make a proper comparison.
>
> (2) We have added two new datasets and architectures to address concerns about the lack of experimental validation. We pushed some of the analysis to the Appendix due to length constraints. We would like to point out Appendix B (Effect of Sparsity Distribution on Other Methods) , C (Effect of Momentum Coefficient for SNFS), D (Existence of Lottery Tickets), E (Effect of Update Schedules), F (Alternative Update Schedules) and H (Additional Plots and Experiments for CIFAR-10) for extended analysis of our results. We have an explanation based on a Taylor series approximation why magnitude pruning and largest gradient growth is optimal (at least greedily) in Appendix F (now Appendix A). Unfortunately we had originally forgotten to mention this in the main text, but we have now updated the last paragraph of Section 4 with a short discussion and a pointer to the appendix.
>
> (3) Regarding SBP (we assume you mean “Structured Bayesian Pruning” https://arxiv.org/abs/1705.07283), VIB (assuming “Variational Information Bottleneck” https://arxiv.org/pdf/1802.10399.pdf), and L0-regularization (https://arxiv.org/pdf/1712.01312.pdf) - we note that these techniques function as an architecture search over channel count distributions and are somewhat orthogonal to our own work. SBP and VIB do _at least_ as much work during training as training the large model they start with (often quite a bit more as there are non-trivial overheads associated with these techniques).  The resulting architectures can be trained from scratch (https://arxiv.org/abs/1810.05270).  Once these techniques find an improved architecture it could be trained with _weight_ sparsity using our technique.
>
>   L0 does slightly reduce the cost during training, but the difference is not very large.  According to Figure 4 of their paper, the cost is decreased from ~3.5 to 3.25 (e11) flops by the end of training, or a reduction of 7%.  The total decrease over the course of training would be even less.

---

> > ### Comment · AnonReviewer1 · 2019-11-13
> > **Response regarding DEN**
> >
> > [Yoon et al. 18] reports the performance they obtain on the entire set of classes, by fine-tuning of the network trained in a continual learning manner (See Den-Finetune in Figure 3 of [Yoon et al. 18]). Since the authors also provide the codes and Den-Finetune model trains the network in a standard setup, I believe that it will be easy to compare against it. Since DEN and RigL both increases the network capacity from a sparse network,  I have concerns regarding the novelty of this work over DEN, and thus want to see the experimental comparison over it.

---

> > > ### Author Response · Authors · 2019-11-13
> > > **Code is incomplete**
> > >
> > > Unfortunately, despite stating "We will release our codes [sic] upon acceptance of our paper, for reproduction of the results," the code provided by the authors omits convolutions.  https://github.com/jaehong-yoon93/DEN
> > >
> > > There has been an open issue since August 22nd, 2018 asking for them.  At this point it seems reasonable to assume the authors are not planning on providing this code.
> > >
> > > The lack of this code combined with the lack of necessary detail regarding how many units are added at each step and how long each iteration is trained for in their paper makes a proper comparison difficult, either by re-running their experiments or using data from the paper.
> > >
> > > Additionally, we note that in the section "Timestamped Inference" they say:
> > >
> > > "In both the network expansion and network split procedures, we timestamp each newly added unit j by setting {z}j = t to record the training stage t when it is added to the network, to further prevent semantic drift caused by the introduction of new hidden units. At inference time, each task will only use the parameters that were introduced up to stage t, to prevent the old tasks from using new hidden units added in the training process."
> > >
> > > This is not realistic for image classification - it presumes we know which subset of labels the input will belong to before we try to classify it.
> > >
> > > We continue to believe that this an interesting line of work that could reduce training costs, but we do not believe that the paper is a baseline that we can compare with.

---

> > > > ### Comment · AnonReviewer1 · 2019-11-13
> > > > **Quick response**
> > > >
> > > > I believe that you could compare against DEN-Finetune with multilayer perceptron as the base network, using the codes provided at the git repository you linked. Timestamped inference should not be an issue with the final finetuned DEN (DEN-Finetune) at all.
> > > >
> > > > Since I do not find the idea of starting with a sparse network and growing it up in a bottom-up manner as novel, as it is already done in DEN, without experimental comparison against it I do not believe that the paper has a sufficient novelty or advantage over it.
> > > >
> > > > Thus I will stick to my original rating of weak reject.

---

> > > > > ### Author Response · Authors · 2019-11-15
> > > > > **We have provided all requested experiments.**
> > > > >
> > > > > We hope you find them useful in re-evaluating our work as they show that RigL substantially outperforms DEN.

---

> > > ### Author Response · Authors · 2019-11-15
> > > **Comparison with DEN**
> > >
> > > We re-emphasize that RigL does not “grow” an architecture. The number of neurons and connections are *fixed* throughout training - RigL does NOT increase network capacity during training. Since the reviewer has made this claim multiple times, we would like to briefly summarize our method (Section-3 of our paper) below to make sure there is no confusion:
> > >
> > > (RigL) takes a predefined architecture (i.e. Resnet-50) and randomly sparsifies its *connections* (not neurons). Training is done on this sparse network. Every T steps we change the connectivity of each layer by dropping least magnitude connections and activating the ones that have the highest expected gradient. Note that the number of deactivated connections are same as the activated ones; therefore total number of connections *doesn't grow or change*: resource requirements are fixed throughout the training. This general idea of training sparse networks with dynamic connectivity is done in earlier work (see SET, SNFS, DSR references in our paper) and it is not the novelty of our work. The novelty of our work lies in using the instantaneous gradient information to select which connections to activate.
> > >
> > > ------------Experiments with DEN---------
> > > We checked the DEN-repo carefully. The code includes only a permuted-MNIST training setup which is only mentioned in the appendix of the paper, all the experiments from the main text are missing. Since there is no information on which hyper parameters used, we used the default values. There is also no option for running DEN-finetune experiments; therefore we added this capability. We ran the following experiments:
> > > (1a) Default training. The initial network starts with 31.14% sparsity and 784-312-128-10 units. The network obtained after 10 permuted MNIST task is 784-384-199-10 and has an overall sparsity of 0.1416. We finetuned the resulting model for 20 epochs (3 different runs). The total FLOPS needed during training is roughly 28,200 GFLOPs (Detailed calculation below).
> > > (1b) We trained the same architecture(784-384-199-10) from a random initial point for 20 epochs (Scratch) and 70 epochs (Scratch+).
> > > (1c) We randomly sparsified the first (95%) and second layer(85.5%) of the architecture found by (1a) and trained the architecture using RigL for 70 epochs. We set inital_drop_fraction=0.2, end_step=5e4, update_interval=200 and drop_fraction_annel=cosine.
> > > (2a) We also tried starting DEN training from a much smaller network of 784-100-50-10. This network started with a sparsity of 0.27 and grew into the final architecture of size 784-175-111-10 with an average sparsity 0.0893. After the continual learning setup, the network is finetuned for 20 epochs and the total FLOPs needed during training is approximately 9,920 GFLOPs.
> > > (2b) Similarly we trained the resulting architectures from scratch. For 20 (Scratch) and 70 epochs (Scratch+).
> > > (2c) RigL training. Same settings same as 1c but using architecture found in (2a).
> > >
> > > The results are as follows:
> > >
> > > |   Experiment           |  GFLOPs    |   3 Runs, Top-1 Accuracy   |Average	 |  Sparsity  |
> > > |---------------------------|---------------|------------------------------------|--------------|--------------|
> > > |(1a) DEN-Finetune  |	28,200       | [0.9824, 0.9794, 0.9802]     | 98.06%	 | 0.1416     |
> > > |(1b) Scratch             |	1,370         | [0.9807, 0.979, 0.9811]       | 98.02%	 | 0               |
> > > |(1b) Scratch+           |	4,780         | [0.9853, 0.9852, 0.9812]     | 98.39%	 | 0                |
> > > |(1c) RigL                   | 350            | [0.9836,0.9841,0.9821]        | 98.32%	 | 0.9258      |
> > > |(2a) DEN-Finetune  |9,920         | [0.9791, 0.9781, 0.9783]      | 97.85%	 | 0.0893      |
> > > |(2b) Scratch             |570             | [0.9818,0.9781,0.9806]       | 98.02%	 | 0                |
> > > |(2b) Scratch+           |1,990          | [0.9824,0.9806,0.9808]       | 98.21%	 | 0                |
> > > |(2c) RigL                   |140             | [0.9816,0.9797,0.9809]       | 98.07%	 | 0.9316      |
> > >
> > > Results show the following:
> > > (A) DEN does not achieve meaningful sparsity -- the networks obtained are only 10% sparse -- far too low to be of any practical benefit.
> > >
> > > (B) Using DEN as a pre-training step and then fine tuning the resulting network is not as efficient as training it from scratch. This confirms results of Liu et.al.(https://arxiv.org/abs/1810.05270).
> > >
> > > (C) RigL requires ~100x fewer FLOPs than DEN and gets higher accuracy than DEN-Finetune.
> > > --------------------------------------------------------------------------------------------------------------------------
> > > Code to reproduce DEN-finetune results: https://drive.google.com/file/d/1eb2a_sB_A3dMemN3QkdFmTry1uljtJxo/view?usp=sharing
> > >
> > > Calculation of training FLOPS for DEN is explained in this anonymous doc: https://docs.google.com/document/d/1WkPwtIT-qSn5XamzLT3hDCQuyLd4P5IJ2zKriBdgFAs/edit?usp=sharing

---

> > ### Comment · AnonReviewer1 · 2019-11-13
> > **Response regarding SBP**
> >
> > What I mentioned as necessary was actual experimental comparison against SBP, VIB, and L0-regularization. If the two methods are orthogonal, then I think you should report how combining your approach with the state-of-the-art Bayesian sparsification methods improves their sparsification performances, since they are indeed state-of-the-art sparsification approaches.

---

> > > ### Author Response · Authors · 2019-11-13
> > > **Response**
> > >
> > > It can be difficult to decide how to allocate compute and human resources for experiments.  We hope that the reviewer can agree that in this case,  we can say for certain without even running SBP and VIB that they will do at least as much work during training as pruning*.  That is they will fall on or to the _right_ of the pruning in all of the plots.  Given this we don't think it makes sense to dedicate such limited resources to these experiments.
> > >
> > > *For example, SBP mentions "At the start of the training procedure, we use pre-trained weights for initialization." meaning that the cost of SBP will be to the _right_ of the pruning line since it must include the cost of pre-training.
> > >
> > > L0 might do slightly less work than pruning, but given the data from their paper (approximately 7% less), it does not seem like a significant enough difference, as the results presented in our paper do 75-95% less work.
> > >
> > > Additionally, we believe one key difference between those techniques and ours is that they will never be able to train a larger network than the largest dense network we can currently train (and in fact it will be smaller, since they introduce memory overheads).
> > >
> > > We also note that we have experiments on MobileNet_v2 which is significantly less overparameterized relative to the networks used in those papers.  For example, SBP finds a VGG network with ~1M parameters that matches the original dense performance on CIFAR-10.  Our 75% sparse mobilenet_v2 ERK network needs fewer than 1M parameters to solve a much more difficult problem (ImageNet).
> > >
> > > Finally, we disagree with the nomenclature that "SBP, VIB and L0 (when applied to neurons)" are sparsification approaches.  They start with dense networks, train dense and result in a dense network that can be trained from scratch.  They are an architecture search over channel counts.  We will consider examining how architectures found with these techniques behave when trained with RigL in future work.

---

> > > ### Author Response · Authors · 2019-11-15
> > > **Comparison with SBP, VIB and L0**
> > >
> > >
> > > Comparison with SBP/VIB/L0
> > > ========================
> > >
> > > (updated numbers to fix one big error in the FLOPs calculations - they were originally made with 50,000 images in the training set rather than 60,000 - and a few smaller ones.  We also add the inference costs of and size of each network and the table below which summarizes the results. After training the full network with RigL, we removed the units that have 0 incoming or outgoing edges to obtain the compact architecture.)
> > >
> > > |                  |  Arch                | Sparsity | Train GFLOPs | Inference MFLOPs | Size (bytes)|Error |
> > > |--------------|--------------------|-------------|-------------------|---------------------------|----------------|--------|
> > > | SBP          | 245-160-55-10 | 0             | 13521              | 97.1                          | 195100        | 1.6   |
> > > | SBP-S       | 245-160-55-10| 0              | 2554               | 97.1                           | 195100        | 1.6   |
> > > | VIB            | 97-71-33-10    | 0             | 13521              | 19.12                        | 38696          | 1.6   |
> > > | VIB-S        | 97-71-33-10     | 0             | 523                  | 19.12                        | 38696          | 1.6   |
> > > | L0              | 266-88-33-10  | 0              | 1964               | 53.284                      | 107092        | 1.6   |
> > > | L0-S           | 266-88-33-10  | 0             | 1356               | 53.284                       | 107092        | 1.6   |
> > > | RigL-300   | 455-185-86-10| 90.98%  | 583                  | 18.214                      | 50058          | 1.6   |
> > > | RigL-100   | 471-121-46-10| 89.41%   | 397                 | 13.346                       | 35219         | 1.7   |
> > >
> > > For RigL we run the following experiments:
> > >
> > > (RigL-100) Training the default 784-300-100-10 architecture with 98% sparsity in the first layer and 90% of that (or 88.2%) in the second layer.  This achieves an error rate of 1.6% in 200 epochs. The total FLOPs done during training are 583 GFLOPs.  We note that we need far fewer than 200 epochs to reach this error rate, but we leave training length as 200 to avoid a confounding factor when comparing to the other methods which reports results at 200 epochs.
> > >
> > > The hyperparameters we use are:
> > >
> > > Optimizer: nesterov momentum, lr=0.2, momentum=0.9
> > > Batch size: 100
> > > L2 decay: 1e-4
> > > Maskupdate_begin_step: 0
> > > Maskupdate_end_step: 40000
> > > Maskupdate_frequency: 200
> > > Drop_fraction: 0.2
> > >
> > > SBP and VIB must do, as a lower bound, as much work as training the dense network.  This requires 13,521 GFLOPs.  For L0, as a generous lower bound, we assume the original network size for the first 10 epochs and the final network size for the remaining 190 epochs, this requires 1,964 GFLOPs.
> > >
> > > This means that RigL does significantly less work than any of these approaches when starting from the original 784-300-100-10 network.
> > >
> > > (RigL-160) We also train the smaller network found by SBP 784-160-55-10 with RigL with 96% sparsity in the first layer and 90% of that (or 86.4%) in the second layer. This achieves an error rate of 1.7% in 200 epochs (again far more than necessary). The total FLOPS done during training are 397 GFLOPs.
> > >
> > > The hyperparameters we use are:
> > >
> > > Optimizer: nesterov momentum, lr=0.2, momentum=0.9
> > > Batch size: 100
> > > L2 decay: 1e-4
> > > Maskupdate_begin_step: 0
> > > Maskupdate_end_step: 70000
> > > Maskupdate_frequency: 100
> > > Drop_fraction: 0.2
> > >
> > > Training the SBP network of 245-160-55-10 from scratch would require 2,554 GFLOPs (SBP-S). Training the L0 network of 266-88-33 would require 1,356 GFLOPs (L0-S).  Training the VIB network 97-71-33-10 would require 523 GFLOPs (VIB-S).
> > >
> > > Even on the networks found through bayesian architecture search, RigL still requires fewer flops to achieve an equivalent error rate.  This demonstrates the orthogonality of these approaches.
> > >
> > > The inference cost of each network is:
> > >
> > > RigL-Large: 18.5 MFlops
> > > RigL-Small: 13.5 MFlops
> > > L0: 53.2 MFlops
> > > SBP: 97.1 MFlops
> > > VIB: 19.12 MFlops
> > >
> > > In short, RigL requires _far_ fewer FLOPs when starting from a larger architecture and still requires 1.5-7x fewer flops when starting from an optimized architecture and the architectures it finds require fewer flops than the bayesian approaches.  RigL can still find more efficient architectures when starting from an architecture that has been found by the bayesian approaches.
> > >
> > > -------------------------------------------------------
> > > - SBP, Structured Bayesian Pruning via Log-Normal Multiplicative Noise, https://arxiv.org/abs/1705.07283
> > > - VIB, Compressing Neural Networks using the Variational
> > > Information Bottleneck, https://arxiv.org/pdf/1802.10399.pdf
> > > - L0, LEARNING SPARSE NEURAL NETWORKS
> > > THROUGH L0 REGULARIZATION, https://arxiv.org/pdf/1712.01312.pdf

---

> ### Author Response · Authors · 2019-11-15
> **Requested Results Are Provided**
>
> We would like to thank reviewer 1 for the suggestion to compare with SBP/VIB/L0 and also DEN.  We have done those comparisons with the standard 300-100 MLP on MNIST, since this is setting where most of these papers report results.  After making the necessary modifications to the provided DEN code (ie adding fine tuning), we have also run it in this setting.  The findings strongly support RigL being much more efficient than DEN and and SBP/VIB/L0, and that RigL can further improve the architectures found by SBP/VIB/L0.

---

### Author Response · Authors · 2019-11-12
**Common Response**

We would like to thank the reviewers for their time and insightful comments.  We begin by addressing some common concerns and share some updates:

(1) Concern that there is not enough experimental validation.  To address this we have added:
  - Experiments with Wide ResNet-depth=22-width=2 on CIFAR-10.
  - Experiments using RNNs for character level language modeling on WikiText-103.  This setting approximately mimics the
     one used in “Efficient Neural Audio Synthesis” where a sparse RNN is found to be useful for a Text-To-Speech task, but
     with open source and easily available data.
  - In both of these new settings the finding that RigL outperforms all other sparse training techniques is confirmed.

(2) We found a synchronization issue with our implementation among multiple training replicas. This has slightly altered the results (generally to be more favorable to RigL and SNFS); we have included a detailed discussion in Appendix  J and updated the code attached.

(3) We have added the performance of dense baselines to Figure 2 (left) and 3 (left).

We believe the additional datasets and architectures make a thorough case for the utility of RigL.

Individual responses to the reviews are provided below.

---

### Decision · Program_Chairs · 2019-12-19

**Decision:**

Reject

**Comment:**

A somewhat new approach to growing sparse networks.  Experimental validation is good, focussing on ImageNet and CIFAR-10, plus experiments on language modelling.  Though efficient in computation and storage size, the approach does not have a theoretical foundation.  That does not agree with the intended scope of ICLR.  I strongly suggest the authors submit elsewhere.

---

> ### Author Response · Authors · 2019-12-20
> **Source of intended scope of ICLR?**
>
> We would like to make sure our future submissions fall within the intended scope of ICLR, but we cannot find a source documenting the intended scope.  This makes compliance difficult.  We hope the AC can help resolve the situation by pointing to a written source that makes the intended scope clear.
>
> The call for papers does not seem to mention anything on the subject of "theoretical foundations".  As far as we can tell our submission clearly fell within multiple categories outlined in the call for papers.

---

> ### Author Response · Authors · 2020-01-07
> **Response Required**
>
> It seems critical for the AC to provide a source for the intended scope of ICLR so that future submissions from the community can make sure they fall within said scope.  If there is no written source available for the intended scope of ICLR how can submissions be expected to fall within this scope?

---

> > ### Comment · Area_Chair1 · 2020-01-08
> > **after the meal**
> >
> > The preference of papers with a theoretical basis over papers which are mostly experimental is indeed not officially rooted within the CFP.  The reviewer scores lean the decision towards a reject, and comments include "The method is more like a set of heuristics rather than a principled approach", "they use the gradients ... somewhat intuitive", "the author also provides some intuitive explanation".

---

> > > ### Author Response · Authors · 2020-01-08
> > > **after breakfast**
> > >
> > > To understand then correctly:
> > >
> > > ICLR has "preferences" which are only known to ACs.  One of these preferences is for "papers with a theoretical basis over papers which are mostly experimental".  There may be others.  Submitters are not told in advance of these preferences, but may have their submission rejected because of them.  That does not sound like a system in which I want to participate.  I hope that future versions of ICLR either make all such preferences explicit, or that unwritten preferences are not used as reasons for rejection.
> > >
> > > I also note that Review 1 stopped responding after we provided all the comparisons and additional datasets that they asked for (with the results in favor of our method), despite saying "I temporarily provide the rating of weak reject. I may change my opinion if the authors provide those results during the rebuttal period."  This is precisely the situation where I would hope the AC might intervene beyond looking at raw aggregate scores, especially since the score 6-6-3 places the paper on the borderline.
> > >
> > > I also note the following quotes, while also noting that quotes from reviews are not enough justification for a decision one way or the other:
> > > "Overall, I found the paper to be very clear and of high quality, and thus I find this to be an interest addition to investigations into the lottery ticket hypothesis."
> > > "Overall, this is a nice paper that should be accepted to ICLR."
> > >
> > > I hope the AC learns from this experience and does a more thorough job if tasked with similar responsibilities in the future.
> > >
> > > Regards,
> > > Erich Elsen